# Head Trajectory Diagrams for Gait Symmetry Analysis Using a Single Head-Worn IMU

**DOI:** 10.3390/s21196621

**Published:** 2021-10-05

**Authors:** Tong-Hun Hwang, Alfred O. Effenberg

**Affiliations:** Institute of Sports Science, Leibniz University Hannover, 30167 Hannover, Germany

**Keywords:** gait symmetry analysis, gait symmetry, head-worn sensor, wearable sensor, inertial measurement unit, eye diagram

## Abstract

Gait symmetry analysis plays an important role in the diagnosis and rehabilitation of pathological gait. Recently, wearable devices have also been developed for simple gait analysis solutions. However, measurement in clinical settings can differ from gait in daily life, and simple wearable devices are restricted to a few parameters, providing one-sided trajectories of one arm or leg. Therefore, head-worn devices with sensors (e.g., earbuds) should be considered to analyze gait symmetry because the head sways towards the left and right side depending on steps. This paper proposed new visualization methods using head-worn sensors, able to facilitate gait symmetry analysis outside as well as inside. Data were collected with an inertial measurement unit (IMU) based motion capture system when twelve participants walked on the 400-m running track. From head trajectories on the transverse and frontal plane, three types of diagrams were displayed, and five concepts of parameters were measured for gait symmetry analysis. The mean absolute percentage error (MAPE) of step counting was lower than 0.65%, representing the reliability of measured parameters. The methods enable also left-right step recognition (MAPE ≤ 2.13%). This study can support maintenance and relearning of a balanced healthy gait in various areas with simple and easy-to-use devices.

## 1. Introduction

Gait symmetry is a key concept in the diagnosis of a pathological gait [1] which can cause more serious health problems in a long- and short-term period. Researchers have studied assessment methods of gait symmetry, and these studies have supported gait rehabilitation and management for neurological, muscular, and sensorimotor problems [1], such as postoperative symptoms of hips and knees [2], concussion [3], stroke [4], and Parkinson’s disease [5]. As gait rehabilitation and management require long-term interventions, there is an increasing interest in gait monitoring systems in daily life settings because the measurement in clinical settings has temporal and spatial limitations [6]. For example, a limited time is allowed for gait analysis in clinics because of their schedule. Patients, in addition, walk a short distance or on treadmills for diagnostic measurement—quite different conditions compared to natural gait in real life [6]. Therefore, the demands for the analysis of natural gait in daily life settings have grown.

Combining various sensors, today’s smartphones [7] and wearable devices have provided health monitoring services, including gait parameters available everywhere and anytime. For example, commercial smartwatches inform users of the number of steps and their gait instability that increases the risk of falling [8]. Smart insoles with pressure sensors can analyze changing forces between the foot and the ground, informing users of their gait symmetry [9]. These wearable devices facilitate gait analysis in daily life settings. However, there are also certain limitations. Single sensor-based devices (e.g., smartphones, smartwatches) hardly distinguish between the left and right steps, providing limited information about gait symmetry. Devices using pressure sensors (e.g., smart insoles) are not free from damage or shape changes caused by forces in every step, which can reduce their product lifetime [10].

Recently, studies are also focusing on head-worn devices for gait analysis. For instance, earbuds and smart glasses are of great interest because they are well-known to consumers and provide a highly seamless user experience. Head mount displays (HMDs), in addition, are directly engaged in virtual reality (VR) and augmented reality (AR) applications. The idea of using head-worn devices in gait analysis is supported by literature. It is reported that head acceleration is helpful to analyze gait events and gait patterns [11]. Researchers have succeeded in embedding gait analysis features on ear-worn devices [12,13,14] or head-worn inertial measurement unit (H-IMU) solutions [15]. These sensors have been developed to provide more gait parameters independently.

In terms of gait symmetry analysis, however, single head-worn sensors have not been developed as an independent solution. In these cases, temporal-spatial gait parameters (e.g., step time) have been measured and differences between the odd and even data periods are compared [13]. However, these kinds of sensors need support from additional sensors because of the recognition of left or right steps. Without independent solutions with single head-worn sensors, their advantages, such as cost effectiveness, spatial freedom, and long product lifetime, are not expected in gait symmetry analysis. No previous studies, nevertheless, have been published for independent solutions using single head-worn sensors in gait symmetry analysis. The development of the independent solutions for gait symmetry analysis has recently been required more often due to the increasing number of head-worn devices such as earbuds, smart glasses, and HMDs.

This paper, therefore, starts to develop independent head-worn sensor solutions for gait symmetry analysis. On the one hand, the head is swaying and oscillating during walking. On the other hand, the hip obliquity (the pelvic rotation about the anterior-posterior axis, influencing the trunk lateral movements) is a gait determinant [1]. As the trunk and head are in coordination during walking [16], the head swaying and oscillating trajectory might be used as a gait determinant. In gait analysis, diagrams have been used, such as the sagittal plane joint angles [17], hip-knee angle-angle diagram [18], and butterfly diagram [19]. These analysis methods are, however, normally used with angular kinematics of low limbs or forces between the foot and ground. Thus, these diagrams are not fit for the linear kinematic analysis of the head trajectory and, moreover, a symmetric format of these diagrams is necessary to display the symmetricity of the data. One of the most widely used diagrams to confirm symmetricity is the eye diagram (also called an eye pattern) in digital communications [20]. With the eye diagrams, circuit designers and testers can intuitively notice asymmetric patterns [21], even with their naked eye. The eye diagram helps circuit designers to decide quickly whether they need to modify the circuit design in terms of the size (or the ratio in the size) of *n*-type and *p*-type transistors which are responsible for falling and rising signals, respectively. Secondly, to build modification plans or to decide exact sizes of transistors, they look up more detailed parameters of the eye diagram, such as the eye height, eye width, and eye crossing percentage. Finally, the eye diagram can be used for a proof of the system reliability.

Motivated by the eye diagram in data communication, in this paper, two *eye diagrams* and a *W-diagram* for gait symmetry analysis were proposed for independent solution of single head-worn sensors. In *eye diagrams*, head trajectories during walking are divided into left and right steps, which look rising and falling signals, respectively. In the *W-diagram*, the divided left and right steps look like the left and right ‘U’ in the alphabet double-‘U’ (W), respectively. These proposed diagrams were designed to help practitioners or therapists to quickly notice gait asymmetry, and to plan gait rehabilitation properly. Materials and methods including the overview of the experiment conditions are explained in the next section. Results, discussion, and conclusion also follow.

## 2. Materials and Methods

### 2.1. Participtants and Data Colection

Twelve healthy participants who can walk normally (female: 6, male: 6; age: 29.8 ± 6.8 years; height: 173.3 ± 8.6 cm) were asked to walk 100 steps on the part of a 400-m running track that is for track and field athletics. They are the same participants in our previous study [22]. Their movement data were collected by an IMU-based motion capture system, so that each participant wore 17 IMUs from XSENS MVN Awinda system during walking. A software from XSENS MVN visualized participants’ walking motion via an avatar in 3-D space at 60 frames per second (fps). Head kinematic data of each participant were extracted and analyzed by Python 2.7 in real-time. Foot kinematic data were also used for comparison.

### 2.2. Sensor Placement and Orientations of Coordination Systems

The IMUs are placed on the head, stern, pelvis, shoulders, arms, hands, legs, and feet (Figure 1). To be specific about the head and foot which are mainly monitored in this study, the head IMU is placed on the back of the head, using a flexible headband. The foot IMU is placed on the front of the foot, which was fixed by the shoelace of shoes. The global orientations (origin: ***O****_Global_*) are defined as the north is the *x*-axis, the west is the *y*-axis, and the upward is the *z*-axis, which is given by XSENS system. The local orientations (origin: ***O****_Local_* or ***O′***) are also defined for the proposed diagrams as *x*′ is the direction of the walking vector, *y*′ is the left of the walking direction, and *z* is the upward direction (Figure 1).

### 2.3. Anatomical Planes in Observation

To observe the level of gait symmetry, two anatomical planes were chosen: the transverse and frontal plane. The transverse plane is viewed from above the head or below the foot. The trajectory of the head’s global movement is observed by this plane. In terms of the frontal plane, the observation position is in the front or back side of the human body. In this plane, the lateral and vertical head movement can be observed. On the coordination systems shown in Figure 1, therefore, the transverse plane offers the gait trajectory on the *x*-*y* (or *x*′-*y*′) coordinate system, and the frontal plane provides the observation on the *y*′-*z* coordinate system.

### 2.4. Temporal Alignment: Step Time

In the gait analysis, one of the well-known methods is using sagittal plane joint angles [17] where changing joint angles in a gait cycle are compared. The repeated motions in the walker’s gait cycle are aligned in a unit interval and compared to each other. In this paper, a step time is defined as the unit interval instead of a gait cycle. The step time is measured with the time differences between the peaks of the head vertical positions as shown in Figure 2. In the sagittal plane joint angles diagram, the hip, knee, and ankle angles can be compared to each other directly after the alignment with the unit interval because the angle changes in a certain range (e.g., from −10° to 60° for the knee). Similarly, the *z*-position in Figure 2 is also immediately compared after temporal alignment because data are repeated in a certain range (e.g., 0.06 m). However, the *y*-positions keep increasing or decreasing depending on the walking directions as shown in Figure 2a and b because the *x*- and *y*-positions are continuously changing when people walk in daily life settings. In this case, it is difficult to compare each trajectory on the *x*- and *y*-axis, so that spatial alignment is also needed.

### 2.5. Spatial Alignment and Walking Vector

In Figure 3, it is clearly observable that both *x*- and *y*-positions are diverse along the walking direction. For example, the P_A_ moves from (4.4, 9.9) to (6.4, 12.8) and P_B_ moves from (−8.6, −33.1) to (−7.8, −36.9). These participants were located in different absolute positions and moved in different directions, and thereby data are not comparable. To compare movements in each step, the *x*- and *y*-positions must be aligned with the geometric transformation to a new *x*′-*y*′ coordinate system. For the new coordinate system, a walking vector is defined between positions (***O****_n_*) as shown in Figure 4. The walking vector is computed as below:(1)$$W_{n}=O_{n}-O_{n-1}=\left(\frac{x_{n}}{y_{n}}\right)-\left(\frac{x_{n-1}}{y_{n-1}}\right)$$
where ***W****_n_* is the walking vector for the *n*-th step (*n* > 1). The position ***O****_n_* and ***O****_n_*_-1_ are defined on the *x*-*y* coordinate system when the *n*-th and (*n*−1)-th head vertical peak are detected. After ***O****_n_* is detected, *n*-th the head trajectory between ***O****_n_* and ***O****_n_*_+1_ are defined on the transverse plane as ***D****_n_* (Figure 4a). For the spatial alignment, ***W****_n_* and ***D****_n_* are transformed to ***W***′*_n_* and ***D***′*_n_* together on the new *x*′-*y*′ coordinate system as shown in Figure 4b. When it comes to six steps with six walking vectors (Figure 4c), unit trajectories are overlapped as shown in Figure 4d. We call the graph of Figure 4d *gait eye diagram at the head type-I* (*GE-H I*).

Data in Figure 4d are divided into positive and negative values on the *y*′-axis. When the head is moving toward the left by stepping with the left foot, *y*′-values are positive. When the head is moving toward the right to step with the right foot, *y*′-values are negative. Normally, heel strike (HS) appears in the middle of the nearest two vertical peaks (VPs) which means the middle of the unit trajectory (***D****_n_*). Therefore, it is easily recognized which foot is stepping on the ground at HS, referring to the sign of the *y*′-values. From Figure 4d, the step lengths and widths are also compared. The length of ***O***′*_n_* (on the *x*′-axis) informs the step length, which can show the step length skew (differences between two feet) and step length jitter (differences of the same foot). In terms of the step width, the positive and negative peaks of the *y*′-value can indicate the head lateral movement, and thereby the symmetricity of left and right step widths can be estimated. In addition, the temporal skew and jitter are also measured and compared. The step time can be calculated by multiplying the frame interval time (16.7 ms at 60 fps) and the number of data points in a step.

When the gait velocity is changed, unit trajectories are mixed up because gait lengths and widths are also changed. Therefore, the length of unit trajectories in the *x*′-axis are normalized between 0.0 and 1.0 as shown in Figure 5, thereby being able to maintain the eye shape, which is visually informative. This diagram is called *gait eye diagram at the head type-II* (*GE-H II*). Although the step length information is lost, the gait symmetry can be analyzed based on unit trajectories.

In Figure 5, one of the most important features is the positive and negative peak. We call them lateral peaks (LPs) because these peaks are from the lateral movement of the head. From the average of LPs (ALPs), the range of lateral movement of the head is analyzed. The second important feature is the average endpoint (AE), which contains information about the lateral bias when the head position is vertically peaked. The average start point (AS) should always be (0, 0), which shows if every unit trajectory is well aligned or not. In this diagram, the average vertical valley (AVV) is also labeled, which can be detected in Figure 6, but not in Figure 5. The AVV is labeled for comparison with other features.

These four features (ALP, AE, AS, and AVV) are also shown in Figure 6. The shapes resemble the letter ‘W’, so that we call g*ait W-diagram at the head (GW-H)*. The unit trajectories can be observed on the frontal plane (*y*′-*z* coordinate system), so that gait symmetry of the vertical movement can be analyzed. The positive direction on the *y*′-axis is for the left side of the head movement, and the negative direction is for the right side. The average vertical valley (AVV) is the most important feature in Figure 6 because it obviously informs about gait symmetry patterns. In Figure 6, AS is the second important feature, showing vertical differences (on the *z*-axis) between AS for the left and right side. Two other features, ALP and AE, are also important; however, it is not visually clear, so it is better to be analyzed in Figure 5. Actually, AE is the same value as the opposite side of AS.

### 2.6. Parameters in the Concepts of the Eye Diagram

In data communications, the eye diagram is much more informative with supportive parameters. The parameters indicate not only symmetricity, but also stability. In this study, new parameters are defined referring to concepts in parameters of the eye diagram: eye height, jitter, bandwidth, signal to noise ratio (SNR). Those parameters are measured at four events at VP, ALP, AVV, or the heel strike (HS) detected by the head motions. For the benchmark, foot sensors are also used to measure comparable parameters to other parameters using head sensors. These parameters are labeled as FT.

#### 2.6.1. Eye Height

The eye height is the distance between positive and negative values in Figure 5. The most noticeable example is the eye height at ALP (EH.LP), which is the distance between the positive peak for the left side (ALP.L.*y*′) and the negative peak for the right side (ALP.R.*y*′). Thus, the equation is as below:EH.LP = ALP.L.*y*′ − ALP.R.*y′*.(2)

The other events (*event*) are also calculated with the distance between values of the left and right side, described as below:EH.*event* = *event*.L.*y*′ − *event*.R.*y*′.(3)

Thus, EH.VV and EH.HS are additionally defined. These values are supported by other diagrams or algorithms. First of all, EH.VV should be supported by Figure 6. For EH.HS, it should be supported by the algorithm of Hwang et. al. (2018) [15], which can detect heel strike (HS) events by analyzing head motions.

In this concept, a comparable parameter using foot motions is the step width (SW.FT). While EH.LP, EH.VV, and EH.HS can be obtained by only head motions, foot positions are required to measure the step width. Due to the outside condition, foot positions are also transformed by using the head’s walking vector (*W_n_*) in Figure 4, and foot position data are also aligned after divided into unit trajectories. After these transformations, left foot motions have positive *y*′-values and right foot motions are on the negative part of the *y*′-axis. The step width was measured as a distance between positive and negative values in foot lateral motions at the head’s vertical valley (VV) event.

#### 2.6.2. Jitter

The concept of the jitter in the eye diagram is the variation of repeated signals in time in data communications. When the jitter is out of the normal range, it means that the system cannot reliably transmit the signals because of fluctuating bandwidth. The jitter, thus, can indicate the reliability of the transmission systems.

In terms of gait symmetry analysis, unit trajectories of the head are also varying temporally and spatially. These variations can be expressed by standard deviation, so that the standard deviation of the step length and step time are used for the spatial and temporal jitter, respectively. These parameters cannot be observed directly in the diagram; however, they can be estimated by the thickness of the overlapped trajectory. Similar in data communications, the jitter in gait symmetry analysis can indicate the reliability of one’s gait.

In total, four methods were used to measure the step length and step time. Three methods were used with head displacement at every vertical peak (SL.VP and ST.VP), vertical valley (SL.VV and ST.VV), and heel strike (SL.HS and ST.HS). The other method used foot kinematics, by measuring the displacement between the nearest two footprints at HS events, as well as their duration (SL.FT and ST.FT).

#### 2.6.3. Bandwidth

The bandwidth in data communications is the upper limit of data transfer rate (also called data frequency). The eye diagram is used to check if the system is available to transmit the signals at a certain data frequency, by analyzing parameters (e.g., eye height, jitter, and signal to noise ratio). In the eye diagram in gait symmetry analysis, the data transfer rate can be replaced by the gait velocity. Changing shapes of eye diagrams at each gait velocity can help people to find the gait velocity range suitable for themselves. In addition, differences between velocities of both the left and right side can provide information about gait symmetry.

#### 2.6.4. Signal-to-Noise Ratio

The signal-to-noise ratio (SNR) is a log scale parameter to display also reliability of data transmission in a digital system. This is a log scale ratio between the signal and white noise power of the system. When the noise power is higher, the signal-to-noise ratio is lower, which reduces the possibility to transmit digital symbols (‘1’ or ‘0’) correctly. This parameter is also related to the eye height and jitter as a signal and noise, respectively. Therefore, the SNR for gait symmetry analysis is defined as below:SNR.*event* = EH.*event*/σ.*event*.*y′*.(4)

The signal-to-noise ratio at an event (SNR.*event*) is obtained from eye height at the event (EH.*event*) divided by the standard deviation in *y′*-value (σ.*event.y′*).

#### 2.6.5. Vertical Movement

In Figure 6, the vertical movement of the head is well displayed. Some features can be parameterized in terms of gait symmetry. First of all, it is obviously observed that the height of AVV is different on the left and right side of the diagram. In other words, the head oscillation is different depending on the stepping foot (left or right). The head oscillation can be measured as the distance between vertical peak (VP) and vertical valley (VV). While each side has one vertical valley (VV), there are two vertical peaks (VPs) at AS and AE. Thus, two parameters can be defined: the vertical distance between at AS and VV (S-VV) and between at VV and AE (VV-E) as below:S-VV = AS.*z* − VV.*z*.(5)
VV-E = AE.z − VV.z.(6)

Secondly, the distance between at LP and VV can be observed, so that the equation is as below:LP-VV = LP.*z* − VV.*z*.(7)

### 2.7. Gait Symmetry Indices

To measure the symmetricity, four coefficients are frequently used: ratio index (RI) [23,24,25], symmetry index (SI) [25,26], gait asymmetry (GA) [27], and symmetry angle (SA) [28]. As the angle is not measured, only three coefficients, RI, SI, and GA were chosen for gait symmetry analysis in this study. The equations of RI, SI, and GA are as described below:(8)$$\mathrm{RI}\left(\%\right)=\left(1-\frac{X_{R}}{X_{L}}\right)\times 100\%,$$
(9)$$\mathrm{SI}\left(\%\right)=\frac{\left|X_{L}-X_{R}\right|}{0.5\cdot \left(X_{L}+X_{R}\right)}\times 100\%,$$
(10)$$\mathrm{GA}\left(\%\right)=\ln\frac{X_{R}}{X_{L}}\times 100\%$$
where *X_L_* and *X_R_* are the absolute values of parameters of the left and right side, respectively. For RI, the denominator (*X_L_*) is higher than the numerator (*X_R_*), meaning *X_L_* ≥ *X_R_*. Otherwise, if *X_L_* < *X_R_*, *X_L_* and *X_R_* become the numerator and denominator, respectively. In terms of original GA [27], the sign of the results can be information about which side is bigger. In this study, however, the absolute value of GA was taken for comparison with two other coefficients which are always positive.

In the measurement of indices for the EH, each side of *y*′-values is taken (*X_L_*: ALP.L. *y*′; *X_R_*: ALP.R. *y*′). In the calculation of the jitter, *X_L_* is σ.*event*.L. *y*′, and *X_R_* is σ.*event*.R. *y*′. For SNR, each side of EH and jitter.

## 3. Results

Figure 7 shows the gait eye pattern (x′-y′; GE-H I) normalized eye pattern (x′′-y′; GE-H II) and gait W-patterns (y′-z; GW-H) of 12 participants. From spatial values (x′, y′, and z) in the diagrams, symmetricity is readily observed in the prior-anterior, lateral, and vertical direction. For example, P2, P5, P9, and P11 are laterally asymmetric patterns, as shown in their GE-H I and GE-H II. Vertical asymmetric patterns can be found as well in P1, P7, P8, P9, P10, P11, and P12 of GW-H. Different shapes of left and right wings and the alignment at the beginning of unit trajectories (*y*′ = 0) can be easily recognized. In addition, GE-H II can illustrate when and how an asymmetric pattern occurs. For example, the first half of P2’s diagram is symmetric, but the second half is asymmetric. In GE-H II, the left side of P7 is more unstable than the right side, and vice versa for P8.

### 3.1. Concept: Eye Height

One of the most interesting features of the eye diagram is the eye height, which is related to the distance of the upper (left) and lower (right) trajectories. Figure 8a is the average eye heights measured by four events. The eye height at the lateral peak (EH.LP), vertical valley (EH.VV), and heel strike (EH.HS) were measured with *y*′. In addition, as a comparable gait parameter, the step width was measured by using foot sensors (SW.FT).

In most cases observed in the measured data, the gait event sequence in a gait cycle is HS, VV, and LP, so that eye heights increase gradually from EH.HS to EH.LP as shown in Figure 8a. In the analysis with Pearson correlation, EH.VV was significantly correlated with EH.LP (significant level (2-tailed): α = 0.05), EH.HS (α = 0.01), and SW.FT (α = 0.05). Nevertheless, all methods showed a tendency of correlation coefficient (ρ > 0.5, *p* < 0.1) except for the coefficient between EH.HS and SW.FT (ρ = 0.290). However, in Tukey’s honestly significantly difference (HSD) as a post hoc test, the average of EH.LP is significantly different from EH.HS (*p* < 0.05) and SW.FT (*p* < 0.05).

In Figure 8b, the three gait symmetry indices (RI, SI, and GA) of each method are compared. These indices of EH.LP and EH.VV are significantly correlated (α = 0.01). The results also showed significant correlations of GA between EH.LP and SW.FT, as well as between EH.VV and SW.FT.

### 3.2. Concept: Jitter

Another main parameter in the eye diagram is the jitter. The spatial and temporal jitter are considered. The spatial jitter for gait analysis is related to the step length, which is measured by four different methods. Three methods are measured with head displacement at every vertical peak (SL.VP), vertical valley (SL.VV), and heel strike (SL.HS). The other method is measured by the distance between the nearest two footprints at HS events (SL.FT). Only SL.VP can be observed directly in GE-H I, which is to ***O****_n_* (see also Figure 4d). The other three methods are measured for comparison. In all four methods, step lengths are correlated to each other (α = 0.05) as shown in Figure 9a. No significant differences were observed from ANOVA analysis and Tukey’s HSD. In Figure 9b, gait symmetry indices of four methods are compared, and all indices of SL.VV and SL.FT are significantly correlated (α = 0.05). Other indices, however, resulted in no significant correlations and differences.

For the temporal jitter, step time was calculated by multiplying the sampled point in one step and 16.7 ms (frame interval at 60 fps). Figure 10a shows the average of step times measured by four methods. The average step times were measured between the nearest vertical peaks (ST.VP), between nearest vertical valleys (ST.VV), and between heel strikes (ST.HS), which are measured at head motion events in a step. The fourth value was the step time measured by foot motion events (ST.FT). They are all significantly correlated in α = 0.01 level. In Figure 10b, three gait symmetry indices of six parameters are compared. The first four parameters are the indices of four step times (ST.VP, ST.VV, ST.HS, and ST.FT). The two other parameters are *x*′′-values at VV (GE_II.VV) and LP (GE_II.LP) in GE-H II (Figure 5). The first four parameters cannot be observed in any diagrams, whereas the last two parameters can be observed in GE-H II and compared with the measured step times. All indices in ST.VP and ST.HS are significantly correlated (RI: α = 0.01; SI, GA: α = 0.05). All three indices in ST.VV and GE_II.VV are also significantly correlated (RI, SI, GA: α = 0.01). Analysis of ANOVA and Tukey’s HSD test revealed that no indices are significantly different; however, RIs of ST.VV and ST.FT show a tendency of difference (*p* = 0.06). 

### 3.3. Concept: Bandwidth

The gait parameter corresponding to the bandwidth in data communications is gait velocity, which is obtained from step length (SL) divided by the step time (ST). These gait velocities are compared in four methods as shown in Figure 11: the average gait velocity at vertical peaks (Vel.VP), at vertical valleys (Vel.VV), at heel strikes with the head (Vel.HS), and at HS with the foot (Vel.FT). Pearson correlation results reveal that Vel.HS is significantly correlated with Vel.VP (α = 0.05), Vel.VV (α = 0.01) and Vel.FT (α = 0.01), as well as Vel.VP and Vel.FT are significantly correlated at α = 0.01 level. In ANOVA analysis and Tukey’s HSD, there is no significant difference. The average gait velocities at each side (left, right) are also measured and compared with gait symmetry indices. In Figure 11b, all three gait symmetry indices of Vel.VV and Vel.HS are significantly correlated at α = 0.01 level. Mean differences (ANOVA, Tukey’s HSD) show no significant results.

### 3.4. Concept: Signal-to-Noise Ratio

The signal-to-noise ratio (SNR) in gait eye diagrams is calculated by the average eye height divided by its standard deviation at vertical peak (SNR.VP), vertical valleys (SNR.VV), and heel strikes (SNR.HS). Additionally, using foot motions, the average step width is divided by its standard deviation, resulting in an SNR (SNR.FT). Figure 12a displays a significant correlation between SNR.VP, SNR.VV, and SNR.HS (α = 0.01). No significant mean differences result, according to ANOVA and Tukey’s HSD test. In Figure 12b, the gait symmetry indices of those four methods result in no significant correlation and mean differences. The only observed fact is that their median values are in the range between around 20 and 40.

### 3.5. Vertical Movement in Gait W-Diagram

Asymmetric vertical patterns are also compared in three parameters as shown in Figure 13: the vertical displacement from AS to VV (S-VV) and from VV to AE (VV-E), as well as between VV and LP (VV-LP). The correlation coefficient of the average S-VV and VV-E is exactly correlated at 1.0 because the left start point is the same as the right endpoint and vice versa. However, LP-VV is significantly different from S-VV and VV-E (both *p* < 0.05) according to Tukey’s HSD. In terms of gait symmetry indices, all three indices of S-VV are significantly correlated with VV-E (α = 0.05) and LP-VV (α = 0.01). With Tukey’s HSD, significant differences are observed between all indices of LP-VV and S-VV (*p* < 0.01), as well as between LP-VV and VV-E (*p* < 0.01).

### 3.6. Validation

All proposed parameters with four methods are measured at head motion events during walking: vertical peak (VP), vertical valley (VV), and heel strike (HS). The prerequisite of the measurement is the detection of those events. These events should occur once in every step, so that accuracy of step counting is chosen as an indicator of the system validation. Table 1 describes a comparison of step counting at VP, VV, and HS, which is obtained by the H-IMU system. Step counting at HS is validated only for both feet conditions [22]. The ground truth is also compared, which was manually obtained by human observers. They monitored walking avatars in the 3-D virtual space and foot kinematic data recorded by XSENS Awinda system that was validated [29]. Mean absolute percentage errors (MAPE) were also measured for both left and right steps. In both, step counting at HS is the highest; however, the mean values are not significantly different. On the other hand, left and right step counting at VV is highest. One of the reasons for this is that double vertical valleys occur frequently. To avoid these events, the detecting algorithm should choose one of those valleys at VP. Although this algorithm reduces errors at VV, it is dependent on errors at VP. For example, when a step is missing at VP, one step is missing at VV as well. Thus, the error rate at VV is the highest. The mean absolute percentage errors at VP are higher than at HS because ground truth is measured at heel strike, so that the range of left and right steps at VP are differently defined from the ground truth. Nevertheless, the MAPE and steps are not significantly different.

## 4. Discussion

In this paper, the main concepts of the eye diagram normally used in data communications were applied to gait symmetry analysis. With this new use of these diagrams, practitioners, therapists, and researchers would be immediately informed about the level of gait symmetry. Easily readable diagrams would play an important role in applications for advanced analysis using high-performance computing systems, as well as simple analysis using portable devices. Today, high-performance gait analysis systems and devices can compare complicated data sets and even implement machine learning algorithms [30]. However, processor units require more power to run complicated methods or machine learning algorithms, and also need processing time and memory space. On the other hand, the simplified diagrams can offer real time feedback because they require devices only to run simple computations, resulting in fast responses. In addition, non-experts in gait analysis can immediately assess human gait symmetry of their own gait as well as others’ gait.

In terms of methods, as a pilot study, twelve participants were involved, by wearing an IMU-based motion capture system (XSENS). Nevertheless, data were valuable because it can be found that parameters are reliably in a certain range and trends in terms of straight gait under the flat ground condition. It is also remarkable that their shapes of head trajectory are all different, which could be one source of personal identification using gait patterns. 

Although SNR in data communications has a unit of decibel (dB) as a log scale parameter, the SNR concept in gait analysis does not result in a log scale value in this study. The first reason is to reduce unnecessary computations. Secondly, the final goal is to compare left and right values, so that it was not important if the parameter is a log scale or not. The comparison results might be different in different groups and conditions. In future studies, it can be figured out if the log scale is more informative or not.

In addition, the eye crossing percentage in data communications plays an important role in the analysis of signal symmetry (e.g., rising and falling signals). The concept of eye crossing percentage is, however, not mentioned in this study because the gait symmetry indices of the eye height can replace it to show the level of gait symmetry. In further research with different conditions, the concept can be additionally proved, by measuring the distance between AEs of the left (positive) and right (negative) side.

When these proposed simple diagrams are realized by head-worn devices (e.g., earbuds, HMD), more participants in various groups can be involved in gait analysis research because those devices are easy to set up on participants. For example, those devices can be easily applied to people with Parkinson’s diseases, post-stroke, or post-concussion, for whom it is currently still very difficult and problematic to perform measurements with elaborate diagnostic systems. The performance of athletes can be also easily diagnosed outdoor without any spatial limitations. For further applications, more studies with various groups of people and conditions are needed with different gait velocities and ground conditions (e.g., slope, curve, softness).

Moreover, the diagrams can provide not only visualization, but also sonification [31]. The trajectories of diagrams are so intuitive that users might easily understand the sound generated by their gait patterns. Gait sonification is one of the emerging methods in the rehabilitation of post-operative gait problems (e.g., hip replacement [2]) or other neuro-structural problems such as Parkinson’s disease [32]. It is also expected that the diagrams might be applied to other body parts (e.g., pelvis, foot) instead of the head during walking.

## 5. Conclusions

This study was undertaken to analyze gait symmetry on head trajectory diagrams using head-worn IMU to inform about gait symmetry. Based on head kinematics, three diagrams, gait eye diagram at the head I and II (GE-H I, GE-H II), and gait W-diagram at the head (GW-H) were designed, which can easily inform about the level of gait symmetry on the transverse and frontal plane. The proposed method using H-IMU implies that today’s head-worn devices (e.g., earbuds, smart glasses, HMD) can easily monitor user’s gait symmetry outside laboratories or clinics. Although this study was conducted with a limited number of participants and conditions, it succeeded in visualizing the status of gait symmetry. In future studies, the visualization methods will be run on head-worn devices independently from the whole motion capture system. Furthermore, if it can provide real time feedback visually or acoustically, it can contribute to rehabilitation and therapy of abnormal gait as on people with Parkinson disease or post-stroke patients, as well as maintenance of users’ healthy symmetry gait in daily life.

## Figures and Tables

**Figure 1 sensors-21-06621-f001:**
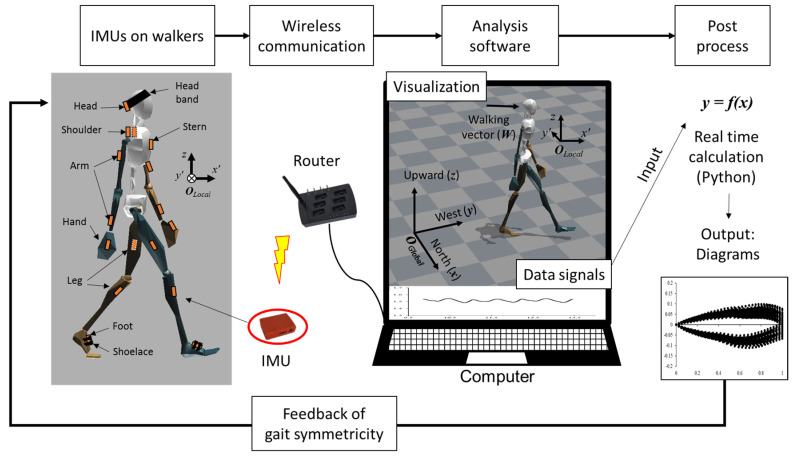
Overview of the proposed real-time feedback system, including the data accusation, the sensor placement, and the orientation of the coordinate systems.

**Figure 2 sensors-21-06621-f002:**
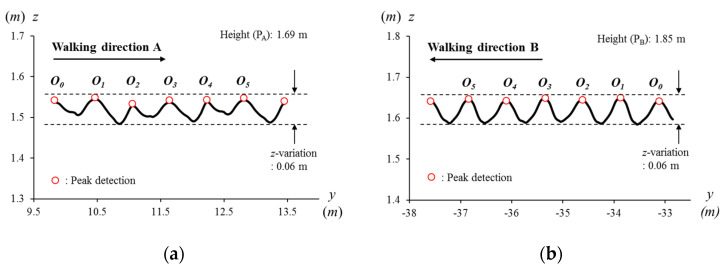
Head trajectories on the *y*-*z* coordinate systems of (**a**) participant A (P_A_) in the walking direction A (***W***_A_) and (**b**) participant B (P_B_) in the walking direction B (***W***_B_). Their *z*-positions are comparable because the positions vary near the walker’s height, independently from walking directions or the global positions.

**Figure 3 sensors-21-06621-f003:**
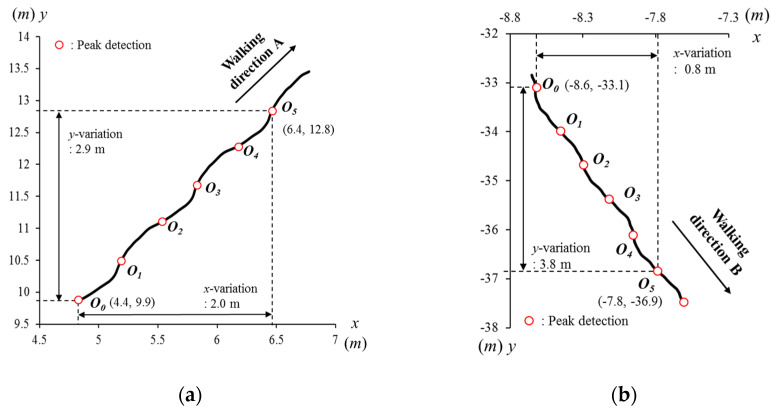
Head trajectories on the *x*-*y* coordinate systems of (**a**) the participant A (P_A_) in the walking direction A (***W***_A_, the north-west) and (**b**) the participant B (P_B_) in the walking direction B (***W***_B_, the north-east). Their *x*- and *y*-positions at peaks are incomparable because the positions have no common reference and also vary depending on walking directions and the global positions.

**Figure 4 sensors-21-06621-f004:**
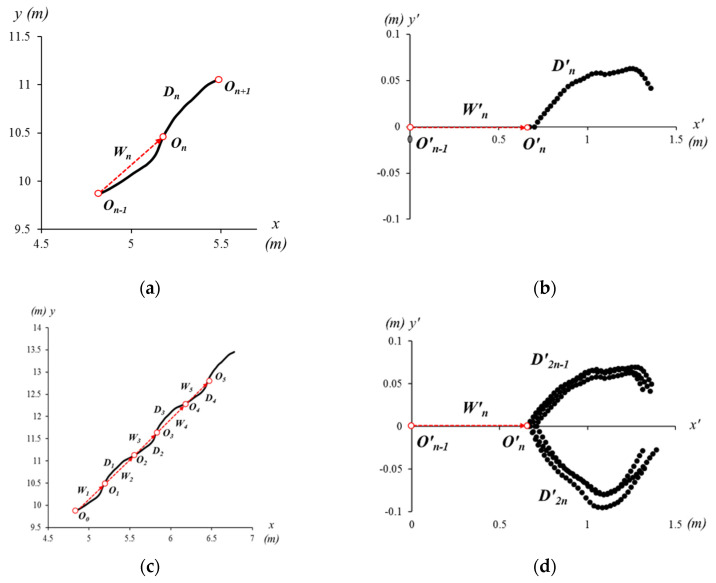
Head trajectories of the participant A (P_A_) for a walking vector (**a**) on the *x*-*y* coordinate systems and (**b**) on the new *x*′-*y*′ coordinate systems, as well as trajectories of six walking vectors of P_A_ (**c**) on the *x*-*y* and (**d**) *x*′-*y*′ coordinate systems (gait eye diagram at the head type-I, GE-H I).

**Figure 5 sensors-21-06621-f005:**
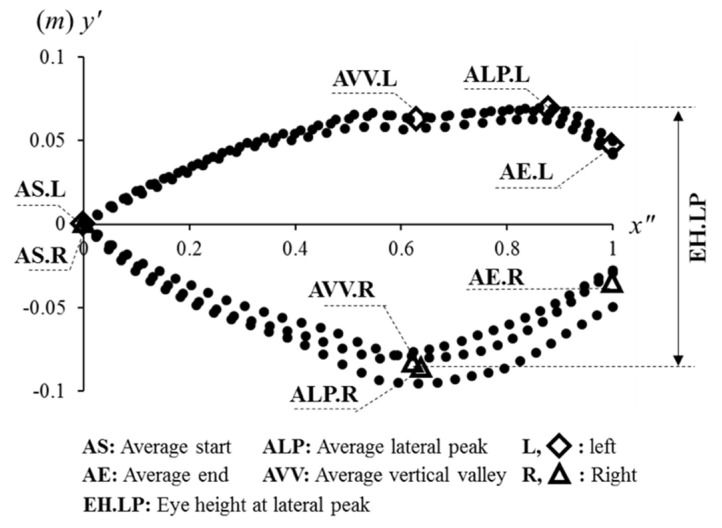
Normalized head unit trajectories of six steps on the transverse plane (*x*″-*y*′ coordinate system; *x*″: 0.0–1.0), also called the gait eye diagram at the head type-II (GE-H II).

**Figure 6 sensors-21-06621-f006:**
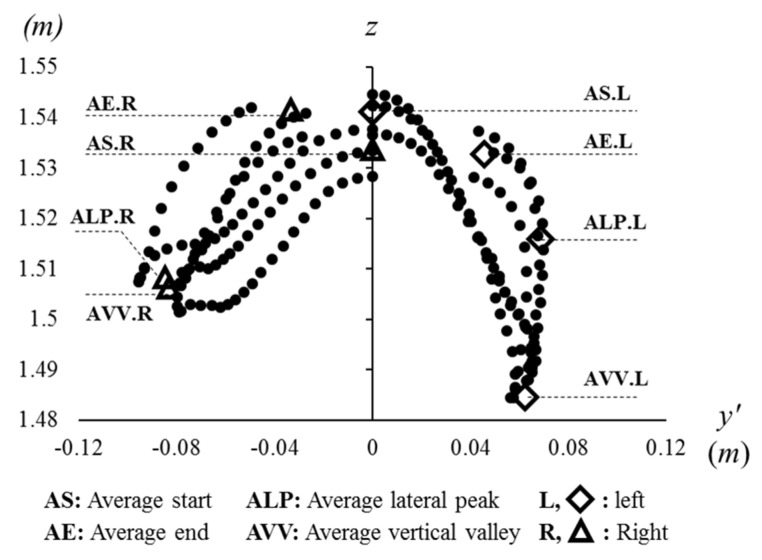
Head trajectory of six steps on the frontal plane (*y*′-*z* coordinate system), also called the gait W-diagram at the head (GW-H).

**Figure 7 sensors-21-06621-f007:**
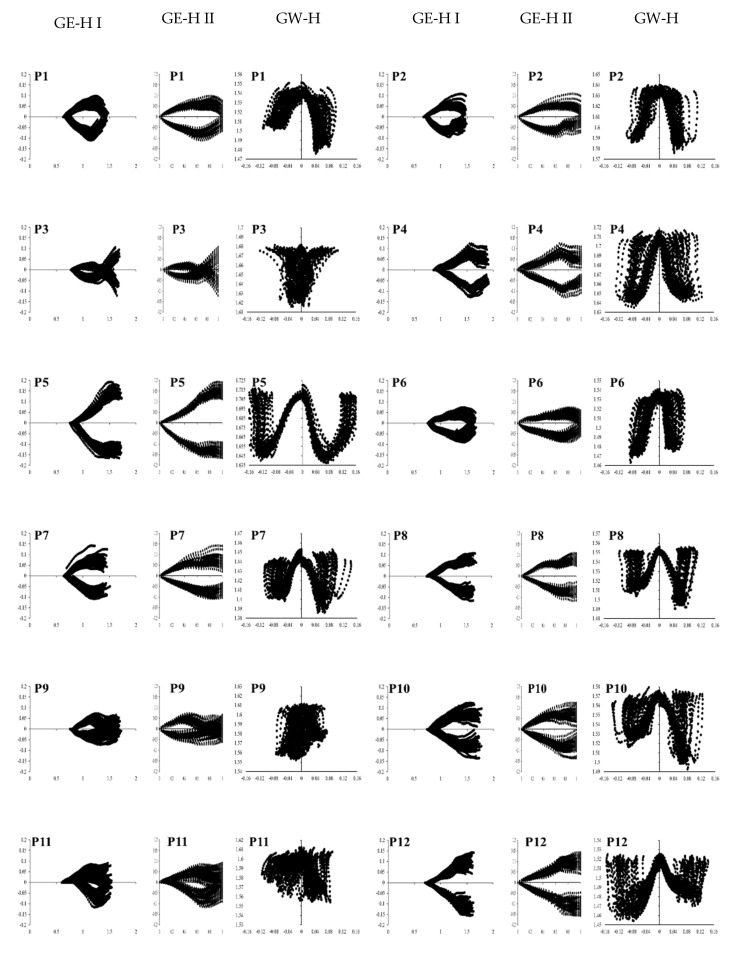
The gait eye diagrams at the head (*x*′-*y*′*,* GE-H I), normalized eye diagrams (*x*′′-*y*′, GE-H II) and the gait W-diagrams (*y*′-*z, GW-H*) of 12 participants (P1–P12).

**Figure 8 sensors-21-06621-f008:**
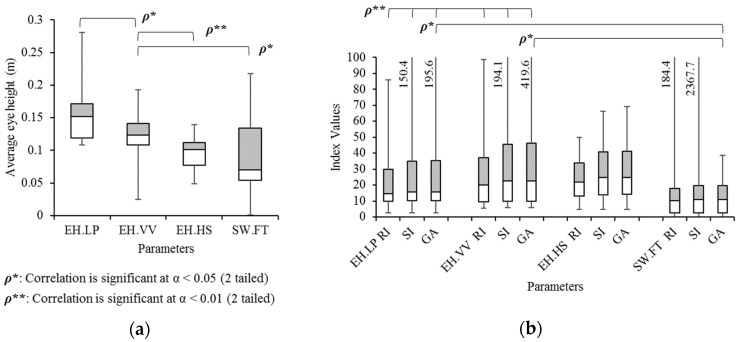
Box and Whisker diagrams of (**a**) the average eye height at three events: lateral peak (EH.LP), vertical valley (EH.VV), and heel strike (EH.HS), and the step width measured by foot sensors (SW.FT), as well as (**b**) their three gait symmetry indices (RI, SI, and GA).

**Figure 9 sensors-21-06621-f009:**
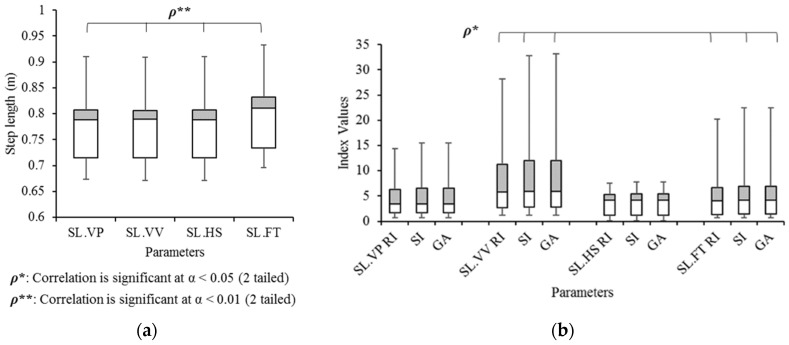
Box and Whisker diagrams of (**a**) the step length in four different measurement sources and (**b**) their three gait symmetry indices.

**Figure 10 sensors-21-06621-f010:**
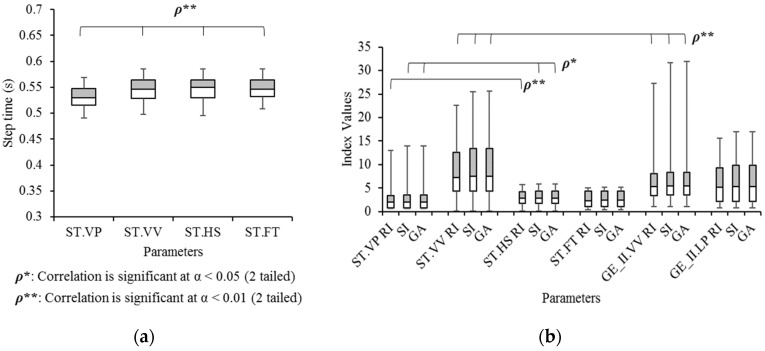
Box and Whisker diagrams of (**a**) the step time in four different measurement sources and (**b**) and their three gait symmetry indices.

**Figure 11 sensors-21-06621-f011:**
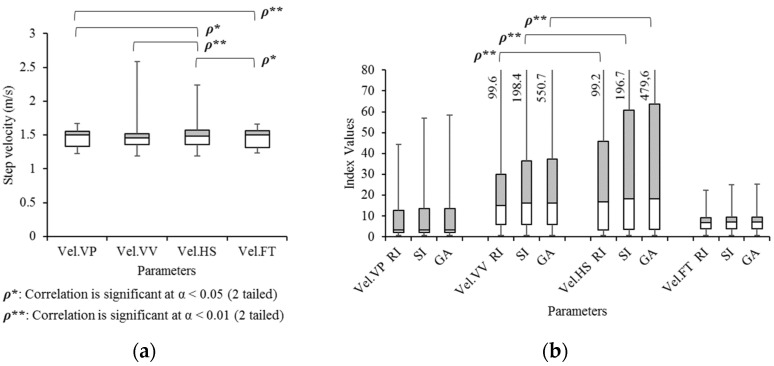
Box and Whisker diagrams of gait velocity (m/s) in (**a**) the average and (**b**) gait symmetry index values in four different measurement methods.

**Figure 12 sensors-21-06621-f012:**
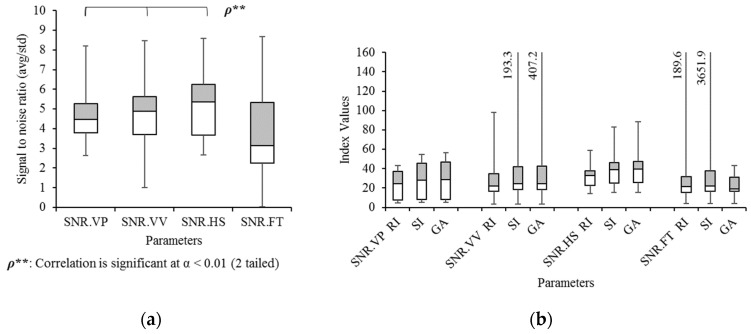
Box and Whisker diagrams of signal to noise ratio (SNR) with (**a**) the average values and (**b**) gait symmetry indices in four different measurement methods.

**Figure 13 sensors-21-06621-f013:**
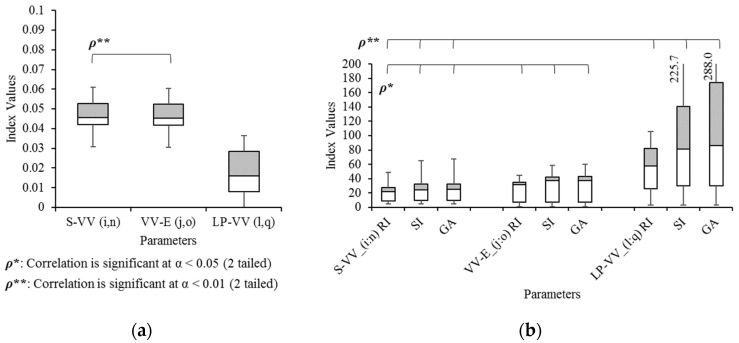
Box and Whisker diagrams of (**a**) the average vertical displacements of the head and (**b**) their gait symmetry indices.

**Table 1 sensors-21-06621-t001:** Comparison of step counting with four methods.

Side	Parameters	VP	VV	HS	Ground Truth
Both	Steps (Mean ± Std.)	85.6 ± 2.9	85.3 ± 3.0	85.0 ± 2.7	85.6 ± 3.1
	^1^ MAPE (%)	0.19	0.48	0.65	
Left	Steps (Mean ± Std.)	42.8 ± 1.8	42.7 ± 1.9	42.3 ± 1.3	42.7 ± 1.4
	^1^ MAPE (%)	0.99	1.59	0.93	
Right	Steps (Mean ± Std.)	42.8 ± 1.2	42.7 ± 1.2	42.8 ± 1.6	42.9 ± 1.8
	^1^ MAPE (%)	1.36	2.13	0.37	

^1^ Mean absolute percentage error.

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
