# Peer review of "Head Trajectory Diagrams for Gait Symmetry Analysis Using a Single Head-Worn IMU"

_sensors, 2021, doi:10.3390/s21196621_

Round 1
Reviewer 1 Report
The authors proposed new visualization methods to measure gait symmetry in human object inertial measurement unit (IMU) based motion capture system. The data presented are solid and compact for the analysis, nevertheless, based on the reviewer's perspective, the research background can be improved further to attract broader audiences.
First, the authors using a commercial IMU sensor from Xsens. The reviewer suggests that the measurement setup can be depicted in a photo or illustration in the manuscript. It can be supporting the readers to get the big picture of how the measurements have been done. (i.e a picture/photo when the patient wears the device and doing the measured activity).
Second, the principle of the system and data acquisition process (real-time sensorimotor feedback systems) should be described more. It can be in the supplementary materials if there is not enough space in the main manuscript.
Third, the patients/persons of the respondents´ information can be described in more detail, such as they are healthy persons or not, with specific diseases or not, the measurements were done after exercise or normal activity, their body weight, and BMI? The reviewer believes these conditions influence the final results.
The reviewer recommends a revision of the proposed manuscript.
Author Response
Dear my reviewer,
I appreciate your review very much. It is really helpful to me. I tried to answer your all questions after “>>>”, and I tried to apply your idea to my manuscript. Nevertheless, some parts would be left, in the case, I tried to explain them. Please find answers and my manuscript. Anyway, thank you again.
The authors proposed new visualization methods to measure gait symmetry in human object inertial measurement unit (IMU) based motion capture system. The data presented are solid and compact for the analysis, nevertheless, based on the reviewer's perspective, the research background can be improved further to attract broader audiences.
First, the authors using a commercial IMU sensor from Xsens. The reviewer suggests that the measurement setup can be depicted in a photo or illustration in the manuscript. It can be supporting the readers to get the big picture of how the measurements have been done. (i.e a picture/photo when the patient wears the device and doing the measured activity).
>>> This is now explained in the new subsection 2.2 with Figure 1 on page 2.
Second, the principle of the system and data acquisition process (real-time sensorimotor feedback systems) should be described more. It can be in the supplementary materials if there is not enough space in the main manuscript.
>>> This is also explained in the new subsection 2.2 with Figure 1 on page 2.
Third, the patients/persons of the respondents´ information can be described in more detail, such as they are healthy persons or not, with specific diseases or not, the measurements were done after exercise or normal activity, their body weight, and BMI? The reviewer believes these conditions influence the final results.
>>> This study is for proposing a methodology, and we did not plan to see correlations between health status and the results. Thus, the health status was not considered in detail. When recruiting participants, we found participants who can walk healthily (normally).
The reviewer recommends a revision of the proposed manuscript.
P.S.: If you prefer, you can download an attachment, which contains the same contents.
Reviewer 2 Report
The paper is well written and presents an interesting investigation and application of state of the art concepts.
My only remark is due to the description of the test environment. The authors should better describe how the sensors are placed and include a picture describing the x,y,z axis in relation to the human testers. Also Dn requires a better description namely with an equation.
Author Response
Dear reviewer,
I appreciate your review very much. It is really helpful to me. I tried to answer your all questions after “>>>”, and I tried to apply your idea to my manuscript. Nevertheless, some parts would be left, in the case, I tried to explain them. Please find answers and my manuscript. Anyway, thank you again.
The paper is well written and presents an interesting investigation and application of state of the art concepts.
My only remark is due to the description of the test environment. The authors should better describe how the sensors are placed and include a picture describing the x,y,z axis in relation to the human testers.
>>> This is explained in subsection 2.2 with Figure 1 on page 2.
Also Dn requires a better description namely with an equation.
>>> I explained Dn again, which is “n-th the head trajectory between On and On+1 are defined on the transverse plane as Dn.” I found that the previous manuscript makes confusion because of the wrong illustration, so I modified Figure 4. I could not add an equation for Dn because it is trajectory recorded by Xsens system. However, I think that readers can understand it better now with the additional explanation and the corrected illustration.
P.S.: If you prefer, you can download the attachment, which contains the same contents.
Reviewer 3 Report
The aim of this paper was to show that head worn devices can be used to assess gait symmetry. The concept of using eye diagrams is introduced.
The results sections (3.6) mentions validation of the methods used and proposed but I am unclear how and how much validation was done. The data collection section mentions that each participant wore 17 IMUs but it is not clear where they were placed and aside from the head mounted one, what they were used for? Was the data from the head IMU validated against data from the others to validate left/right step recognition for example?
Overall, I am still unclear as to the use of the proposed eye diagrams.
Abstract
Line12: should possibly be '..measurement in clinical settings can differ from gait..'
Introduction:
Good overall but some of the text/language is difficult to follow.
Line 44: 'detect the unstable gait' might be better as '..gait instability'
Line 84: should this be- '...notice asymmetric patterns with the naked eye' or similar?
Lines 90-99: This is Methods I think and not needed here?
Materials and methods
Overall, while this section was very long, it was difficult to follow and I am unclear about some of the basic methodology such as IMU placement and use?
I am not clear how step counting was done by human observers? Was this using IMUs on the feet also?
Line 103: am unsure what 'the part of a 400m track and grassy soccer field' means
Lines 101-109: Details needed on IMU placement
Author Response
Dear reviewer,
I appreciate your review very much. It is really helpful to me. I tried to answer your all questions after “>>>”, and I tried to apply your idea to my manuscript. Nevertheless, some parts would be left, in the case, I tried to explain them. Please find answers and my manuscript. Anyway, thank you again.
The results sections (3.6) mentions validation of the methods used and proposed but I am unclear how and how much validation was done.
>>> This study needs to validate the algorithm to divide the head kinematic into each step, automatically. The automatic data rearrangement is important to generate the proposed diagrams. To divide data into each step, algorithms of the step detection are needed, and these algorithms were based on peak detection. Based on peak detection with HS was validated (for both feet) in a previous study. In this study, peak detection with vertical peak (VP), and vertical valley (VV), and heel strike (HS) were compared to the ground truth. The ground truth is obtained by manual step counting from human observers who watched walking avatars in 3-D virtual space generated by Xsens Awinda system (already validated and used by many researchers). In addition, for this study, algorithms should recognize left or right steps to measure the symmetricity of the gait. Thus, step counting of left and right steps were separately measured and compared to the ground truth.
The data collection section mentions that each participant wore 17 IMUs but it is not clear where they were placed and aside from the head mounted one, what they were used for?
>>> This is also explained in the new subsection 2.2 with Figure 1 on page 2.
Was the data from the head IMU validated against data from the others to validate left/right step recognition for example?
>>> Yes, the ground truth data are the data from IMUs attached on the feet. This is manually monitored by human observers who monitored 3-D animations and signals recorded by the motion capture system.
Overall, I am still unclear as to the use of the proposed eye diagrams.
>>> One of the most used diagrams is the eye diagrams in data communication for assessing the symmetricity of signals, thereby confirming the reliability of the systems. The eye diagram helps circuit designers a lot because of its intuitive illustration. Practitioners, therapists, and researchers involved in gait analysis and gait rehabilitation can be supported by the proposed diagrams. It might be easily read by non-experts due to its intuitive structures so that these diagrams can be shown in commercial mobile apps to check their own gait symmetry.
Abstract
Line12: should possibly be '..measurement in clinical settings can differ from gait..'
>>> It was changed to '..measurement in clinical settings can differ from gait..'
Introduction:
Good overall but some of the text/language is difficult to follow.
Line 44: 'detect the unstable gait' might be better as '..gait instability'
>>> It was changed to '..detect gait instability.'
Line 84: should this be- '...notice asymmetric patterns with the naked eye' or similar?
>>> It was changed to ‘...notice asymmetric patterns with the naked eye'
Lines 90-99: This is Methods I think and not needed here?
>>> I deleted the part explaining planes and revise the end of the introduction. Nevertheless, I think it is valuable to briefly explain eye diagrams and W-diagram for the basic knowledge of these diagrams.
Materials and methods
Overall, while this section was very long, it was difficult to follow and I am unclear about some of the basic methodology such as IMU placement and use?
I am not clear how step counting was done by human observers? Was this using IMUs on the feet also?
>>> Xsens system provides an animation with an avatar, as well as signals of feet signals. Figure 1 would help your understanding. Human observers watch the animation and feet signal together. So that they count the number of steps exactly, which can be the ground truth. The Xsens Awinda system and step counting at HS using H-IMU were validated in other places, where I cited in the manuscript.
Line 103: am unsure what 'the part of a 400m track and grassy soccer field' means
>>> 400 m running track for track and field athletics. And the grassy soccer field means the grass field for soccer games. The previous study used gait kinematic data measured on both ground conditions. However, this paper used data only from walking on the running track, so that now I deleted “grassy soccer field”.
Lines 101-109: Details needed on IMU placement
>>> This is also explained in the new subsection 2.2 with Figure 1 on page 2.
If you prefer, you can download an attachment, which contains the same contents.
Round 2
Reviewer 1 Report
The revised manuscript shows a significant improvement. All the reviewer´s suggestions have been done in the manuscript revision, also the questions have been answered reasonably.
The revised manuscript can be accepted for the publication in the MDPI Sensors.